# Investigating Food Packaging Elements from a Consumer’s Perspective

**DOI:** 10.3390/foods9081097

**Published:** 2020-08-11

**Authors:** Ageliki Konstantoglou, Dimitris Folinas, Thomas Fotiadis

**Affiliations:** 1Department of Production and Management Engineering, Democritus University of Thrace, 67100 Xanthi, Greece; angiekonsta@gmail.com (A.K.); dr.fotiadis.thomas@gmail.com (T.F.); 2Department of Supply Chain Management, International Hellenic University, 60100 Katerini, Greece

**Keywords:** packaging, food industry, food supply chain, packaging elements

## Abstract

This study aims to identify and evaluate packaging elements in the food industry, taking into account various business areas/disciplines. The research was conducted with a sample of 1219 customers. An initial pool of 43 packaging items was developed, aiming to examine the elements that have a relationship with consumer behavior in buying food products. Exploratory factor analysis (EFA) was conducted on a random split-half sample of the data to examine the factor structure of these elements in the general population. Confirmatory factor analysis (CFA) was conducted in the holdout sample. The EFA of the packaging items resulted in seven factors: (1) Informational content, (2) Content protection and recognition, (3) Smart functioning, (4) Geometry, (5) Environmental friendliness (6) Endurance, and (7) Coloration. The CFA in the holdout sample supported this factor structure. The findings are informed by the consumer attitudes and predispositions towards packaging, thus having useful managerial applications.

## 1. Introduction

Packaging is one of the most crucial operations in the food industry [1,2]. Almost all of the foods we trade or consume come in some sort of packaging in one form or another. Packaging of food products includes all the materials, of any kind, used to protect, manage, deliver and present products, from raw materials to finished products, from the producer to the user and/or the end-consumer. Apart from the functional role of packaging, there is also the communicative role, quite simply because it becomes the voice and face of the producer’s image and identity [3,4,5].

Food packaging is a critical issue because it ensures that the product remains in good condition and ready for consumption [6]. In other words, the packaging aims to protect the food, not only during transportation and from physical damage, but also from microbial and bacterial contamination/damage, as well as climatic hazards. The role of packaging in increasing consumer awareness of environmental issues is considered of great importance, as well as the adoption by companies in the food industry of concepts such as the use of recyclable, environmentally friendly packaging materials [7,8,9].

Therefore, it is clear that food packaging needs to be holistically considered taking into account its various aspects and disciplines. In the literature, there are many research initiatives that have proposed a packaging design and development framework including a number of factors. As one of the pioneers in food packaging, Paine [10] proposed the following factors: product needs, distribution needs and wants, packaging materials, machinery and production processes, consumer needs and wants, market needs and wants, and environmental performance.

Furthermore, Coles, McDowell and Kirwan [11] proposed a framework for a packaging strategy, as follows: (1) technical requirements of the product and its packaging to ensure pack functionality and product protection/preservation throughout the pack’s shelf life during distribution and storage until its consumption; (2) customer’s valued packaging and product characteristics, for example, aesthetic, flavor, convenience, functional and environmental performance; (3) marketing requirements for packaging and product innovation to establish a distinct (product/service) brand proposition to protect brand integrity and satisfy anticipated demand at an acceptable profit in accordance with a marketing strategy; (4) supply chain considerations such as compatibility with existing pack range and/or manufacturing system; (5) legislation and its operational/financial impacts, for example regulations regarding food hygiene, labelling, weights and measures, food contact materials, due diligence etc.; and finally, (6) environmental requirements or pressures and their impacts, for example light weight materials to reduce the impact of taxes or levies on the amount of packaging used.

Many researchers have identified the multifunctional nature of packaging specifically in the food industry. Rundh [12] points out that: “*in today’s market, packaging consists of three functions which include logistics, commercial and environmental functions*”. Indeed, today more than ever, companies have come to realize that packaging can certainly affect consumers’ decision-making, as well as improve the performance of a business in terms of storage and transport, by standardizing their respective logistics activities, at the same time as minimizing their operational costs and giving the market a pro-environmental image with a high sense of social responsibility [13,14,15,16,17].

Based on the above, this study appreciates also the need for a multi-disciplinary view of packaging in the food industry. This view takes into account marketing, logistics, food technologists and environmental requirements.

The main objective of this study is the identification and categorization of the packaging elements of food products by consumers that refer to all the above disciplines. For this reason, primary data were collected through a research initiative via a questionnaire, which was filled by consumers of food products in four (4) cities in Greece. An initial pool of packaging items was developed, aiming to examine the marketing elements that have a positive relationship to consumer behavior in buying food products. The main research questions of this study are as follows:

RQ1: What do consumers consider the key elements of food packaging?

RQ2: Can we group these elements into factors?

The answers to these research questions of the study can be used by marketing and logistics managers, food technologists and executives responsible for environmental issues in the designing of the packaging on the products. By recognizing the importance, marketing managers, logisticians, food technologists and corporate social responsibility (CSR) and environmental executives ascribe to it; can take into account those elements that are highly appreciated in terms of all of the above components.

The paper is organized as follows: The next section identifies the elements of the packaging of food products from the point of view of marketing, logistics, food technology and the environment. Based on these elements, primary research is applied in the food products market. The sample included responses from consumers of food products. The next section presents and discusses the findings, ending with discussions, conclusions and recommendations in the final section.

## 2. Materials and Methods

Packaging serves various significant roles and goals and in various functional business areas:Marketing, which aims to attract consumers to buy the product. A well-designed package attracts the attention of the consumer and is therefore an effective communication tool and an important tool in product differentiation.Logistics and Supply Chain Management, which supports the physical protection of products during their handling and storage processes, against damage, shock, vibration, temperature, heat, moisture, etc., also including the unitization of foods, sorted from one type of packaging to a bigger load unit, in order to facilitate their movement within the food supply chain.Food technology, which aims to achieve consumer health protection against microbial and bacterial contamination/spoilage, taking into account the climatic hazards of the products, by keeping them healthy, clean, fresh, sterile and safe for their intended shelf life. Moreover, they provide information to consumers on topics concerning the use, consumption, storage, and recycling of packaged foods.The environment, which aims to reduce the impact of packaging on the environment or for the packaging to be produced by using reusable, recyclable and renewable materials.

The above features of packaging are served by a variety of elements, which comprise factors/components that have been identified to serve the requirements of the four aforementioned functional areas. Nancarrow et al. [18] use the term “attributes”. They argue that brands of food products use a range of packaging attributes, combining colors, designs, shapes, symbols, and messages, which collectively make an impact on consumers’ buying behavior. As Silayoi and Speece [19] point out, these attract and sustain attention, helping consumers identify with the images presented. Many researchers also identify a number of elements that increase the efficient and smooth flow of products across the supply chain [20,21] and support the traceability of the food products [22,23]. Furthermore, as Guillard et al. [7] argue, an innovative sustainable form of packaging aims to address food waste and reduction of losses by preserving food quality as well as food safety issues, aiming to prevent food-borne diseases and chemical contamination.

Overall, many researchers have tried to identify the key elements of the package (in general, and especially in the food industry) which have an impact in the four aforementioned disciplines that are involved in the packaging of food products [24,25,26,27,28]. Based on the above studies, as well as the findings of Konstantoglou et al. [29,30,31] concerning research initiatives that have concentrated on the food industry, 43 elements have been extracted, which were then classified into the following four categories (Table 1):Informational elements: information about the company, information about the product (ingredients), nutrition information, production or remaking techniques, quality standards marks, compliance with environmental practices, data that support traceability, lot number, product identification coding schemes such as barcode, QR-code, etc., marks for flammable/hazardous materials, proposed methods of consumption, storage conditions and brand elements (logo, slogan, symbol, etc.).Operational elements: functional (physical) elements, such as protection of the product from theft, protection of the product from moisture, ease of placing/mounting the product on the shelf, does not expose the product to light, allows visual contact with part of the product, ease of transportation and handling, while also permitting packaging in larger packages/logistics units (carton, pallet, etc.).Physical elements: physical characteristics of the packaging, such as size (marginally bigger than the product size), volume (marginally bigger than the product volume), shape (following common/typical shapes e.g., square, rectangle, triangle, circle), and material/components (e.g., made of durable materials, materials that add prestige to the product, materials that are environmentally friendly, materials that can be reused and materials that allow for elongation), being waterproof, withstanding mechanical stress, corrosion and wear, having a low cost (low price of production or recycling), and light/low weight.Visual elements: vivid (strong) colors, only one color (monochrome), only white color (background), with many blank parts and/or product photography/image/graphics.

It should be noted that items falling within the current legislative, regulatory and institutional framework have been excluded, since they are mandatory.

These elements will be applied to achieve the objectives of the present study, which is to evaluate the importance and categorization of the elements from the point of view of consumers of food products.

### Research Method

The research was focused on the food sector and included the examination of the significance of the 43 items presented in the previous section by food consumers. Specifically, 1219 consumers participated in the survey by filling out a questionnaire that initially outlined their demographic characteristics and purchasing behaviour; they were then asked to assess the significance of the packaging elements using a five-point Likert scale (from 1: Not significant to 5: Very significant). Before the questionnaires were administered, they were pilot-tested using the content validity method, and checked for the appropriateness of the elements to have a clear understanding of the questions confirmed by the questionnaire samples. In addition, in order to assess the reliability of the questionnaire, Cronbach’s alpha was estimated. For this purpose, a prototype of 120 pretest questionnaires was taken for pilot testing. The results (Cronbach alpha = 0.926) show that the questionnaire used in this study has a high reliability for achieving its main objectives.

The survey focused on four cities: Athens (the capital of Greece), Thessaloniki (the second largest city in the country), Larissa and Katerini (both large urban centres). The aim was to collect a number of completed questionnaires in proportion to the population of each city. In total, 1219 questionnaires were collected, from 582 (47.74%), 310 (25.43%), 181 (14.85%) and 146 (11.98%) consumers in each city, respectively. The sampling method used in this study was random sampling. The sampling locations were the stores of large retail chains and the research period was between March and June 2019. The collected data were analyzed using SPSS (v.21). The analysis included descriptive and inferential analysis and followed a systematic approach as presented in the following Figure 1:

At first, the large number of observations among consumers (*N* = 1219) allowed the separation of observations into two subgroups with similar numbers, for which the structural analysis was performed in the first subgroup (*N* = 609), and the confirmation of the structure of the model that emerged during the investigation for the second subgroup (*N* = 610). The consumer sample was divided into two groups with the help of the SPSS random number generator. After the separation, each respondent group was statistically independent by gender (x^2^ (1) = 1532, *p* = 0.216), age group category (x^2^ (4) = 5780, *p* = 0.216), place of residence (x^2^) (1) = 3450, *p* = 0.063) and educational level (x^2^ (2) = 2462, *p* = 0.292). The group was also statistically independent by frequency of purchase of packaged food (x^2^ (4) = 3729, *p* = 0.444), information on the food packaging (x^2^ (1) = 2418, *p* = 0.120) and the magnitude of the influence of the packaging for the purchase of the product (x^2^ (4) = 1983, *p* = 0.739)).

The PCA (principal components analysis) method of the main components was applied in the first part of the responses (*N* = 609) to detect the proximity of the questions based on the consumers’ responses. The combination of these two exploratory methods, as well as the qualitative analysis of the concepts described in each question, led to the possible structure of each part of the questionnaire; this structure was confirmed with the data of the second subgroup (*N* = 610).

Initially, exploratory factor analysis was used to identify the appropriate grouping of questions into factors in order to optimize the model’s adaptation to the respective data. The results of the exploratory analysis were used in conjunction with the hierarchical classification of the questions in order to remove certain questions as well as merged factors. In the last step, confirmatory factor analysis was applied to the second part of the sample. Since the questions were distinct, the case for multivariate regression was not supported; therefore, the maximum likelihood (ML) method could not provide reliable calculations of the model’s coefficients or the adjustment indicators. For this reason, the corresponding indicators were calculated with the corresponding robust process of maximum probability, while the DWLS (diagonally weighted least squares) method was also used to control the model; this model adaptation method is more suitable for the case of ordinal variables as in the case of the present research [32,33].

The statistical analysis was performed with the SPSS program while the exploratory (EFA) and confirmatory factor analysis (CFA) were performed using the statistical programming language R equipped with the psych [34,35] packages.

## 3. Results and Discussion

### 3.1. Profile of Sample

Consumer’s demographics are provided in Table 2.

There was no significant difference between gender and place of residence (χ^2^ (1) = 2325, *p* = 0.127), nor between gender and educational level (χ^2^ (2) = 0.353, *p* = 0.838). About 2 in 3 respondents (804, 66%) reported buying packaged foods one or more times per week. 905 consumers (74.2%) stated that they consciously use the food packaging to make a purchase decision, while about 1 in 3 (449, 36.8%) stated that they are highly influenced by the packaging in their decision to purchase the product.

Consumers of the sample appreciate differently the importance of the 43 elements. The results show that the most important packaging items according to consumers (with a threshold of 3.5) per category (as provided in Figure 1) are as follows:Informational: “Provides nutritional information”, “Includes quality standard marks”, “Includes marks that show compliance to environmental practices”, “Includes marks for flammable/hazardous materials storage conditions and brand elements”, “Designates protected origin name”, and “Reports production or reproduction techniques”;Operational: “Protects the product from moisture”, “Does not expose light to solar radiation”, “Can easily be transported and handled”, and “Does not expose the product to light”;Physical: “Does not allow odors to leak”, “Is produced by materials that are environmentally friendly”, and “Is produced by materials that can be reused and materials that allow for elongation”;Visual: the highest mean was “Has a picture” even if it did not pass the threshold.

In general, consumers considered almost all of the elements of the first three categories (informational, operational and physical) important (with higher than average values). The responses focused on the safety and quality of the products and the information provided by the packaging, and less so on the aesthetic elements. Of particular interest are the responses that show the acquisition of the environmentally friendly awareness of consumers.

### 3.2. Exploratory Factor Analysis

Exploratory factor analysis (EFA) was then applied, as was hierarchical classification, for the data of the first consumer subgroup (*N* = 609). This process involved the remaining 35 of the 43 elements, while the other 8 elements were removed as they presented a different pattern of responses from the remaining ones in the same category.

The structure was investigated using the parallel analysis method which calculates the eigenvalues of the main factors for a random sample of the same size as the control data, and then compares the eigenvalues of the factors proposed by principal components analysis (PCA) with the eigenvalues of the resulting factors for the same random sample size (Horn, 1965). From the parallel analysis, 10 main factors with higher eigenvalues than the corresponding main factors of the random sample were indicated. Figure 2 represents the eigenvalues of the two samples (random and control).

This was followed by the identification of the 10 factors using the method of ordinary least squares minimum residuals and oblimin rotation. Non-rectangular rotation was chosen as it is reasonable to assume that the factors describing a package are correlated. Table 3 presents the results of the process, sorted by the load of the questions for each factor. The second column (HCA) shows the position of each question according to the results of the hierarchical classification.

From Table 2 and the comparison of the question loads in the factors, the following changes in the structure of the factors are shown, as indicated by the hierarchical classification:Transfer of inf.q7 (INF1) from MR1 (load 0.29) to MR3 (load 0.28).Consolidation of VIS1 and VIS2 into one VIS factor, which was implemented, since VIS2 had only one question.Consolidation of INF1 and INF2 into one factor, which was not implemented as the two factors had significant conceptual differentiation.Transfer of phys.q6 (PHYS4) from MR4 (load 0.47) to MR7 (load 0.24).Transfer of phys.q8 (PHY1) from MR5 (load 0.47) to MR9 (load 0.18).Transfer of oper.q4 (OPER1) from MR3 (load 0.63) to MR5 (load 0.12).Consolidation of OPER2 (oper.q10) and OPER3 (oper.q8) with OPER1.Transfer of phys.q7 (PHYS4) from MR3 (load 0.8) to MR7 (load 0.13).Transfer of inf.q10 (INF3) from MR7 (load 0.36) to MR10 (load 0.26).Transfer of inf.q11 (INF3) from MR3 (load 0.35) to MR10 (load 0.25) due to special conceptual similarity with inf.q10.

Table 4 presents the effectiveness of the proposed factors as well as the correlations between them.

The elements phys.q8 (load 0.18), oper.q4 (load 0.12) and phys.q7 (load 0.13) were selected for removal, as their load in their new position is extremely small. In addition, it was decided to delete oper.q3: “Easily placed on the shelf” due to its conceptual differentiation from the other two elements in the OPE3 factor. Table 5 presents the coincidence of the predictions of the two methods.

The final model proposed by the combination of hierarchical classification and exploratory structure analysis is as follows:

INF1 = ~ inf.q1 + inf.q2 + inf.q3 + inf.q4        # MR6

INF2 = ~ inf.q5 + inf.q6             # MR6

INF3 = ~ inf.q7 + inf.q8             # MR3

INF4 = ~ inf.q10 + inf.q11            # MR10

OPE1 = ~ oper.q2 + oper.q8 + oper.q10 + oper.q13  # MR5

OPE2 = ~ oper.q5 + oper.q12            # MR8

OPE3 = ~ oper.q11 + oper.q9            # MR4

PHY1 = ~ phys.q1 + phys.q2 + phys.q3      # MR9

PHY2 = ~ phys.q10 + phys.q11 + phys.q14      # MR1

PHY3 = ~ phys.q5 + phys.q6 + phys.q13       # MR7

VIS = ~ vis.q1 + vis.q2 + vis.q3 + vis.q4      # MR2

The final elements that were not involved in the grouping are presented in Table 6.

Table 7 presents the structure of the questionnaire after the exploratory structure analysis together with the internal reliability of each factor.

The presence of factors with an internal reliability of less than 0.6 could not be accepted, so it was decided to merge them with similar factors. An exception was the coloration factor, which could not be merged with any other factor. The final model that participated in the confirmatory structure analysis contained the 31 items in seven groups:Informational content; providing nutritional information, information about the production techniques, the country of origin, quality markings, markings indicating adherence to environmental practices, items that help make the product traceable using auto-id practices, markings to show if it is flammable or contains other dangerous materials, etc.Content protection and recognition; referring to the items that support the protection of the product (from humidity, sun light, etc.) and its recognition by the consumer/user allowing visual contact with part of the product and helping him recognize its contents.Smart functioning; providing specialized capabilities to handling of the product.Geometry, including physical items that refer to geometric attributes of the packaged food regarding its shape, size and volume.Environmentally friendly; including physical items that provide and/or emphasize the environmentally friendly attributes of the product.Endurance; including physical items that strengthen the durability of the product.Coloration; including visual items of the package (such as the use of bright colors, or one color, or having blank spaces.

Table 8 presents the final structure of the questionnaire after the exploratory analysis.

The model that was chosen is illustrated in Figure 3.

### 3.3. Confirmatory Factor Analysis

Table 9 presents the results of the confirmatory structure analysis in the second part of the consumer sample (*N* = 610). As shown, the process of adapting the data to the theoretical model was successful and was completed after 72 repetitions of the process. The model had very good data adjustment (x^2^ (413) = 1070.5, *p* < 0.001), while the adjustment indicators had satisfactory values (CFI = 0.965, TLI = 0.960).

As expected, due to the ordering nature of the variables, the results of the MLR method are not accepted; however, the DWLS method demonstrates good model adaptation for the second part of the data. The corresponding standard coefficients for the sample under consideration are shown in Figure 4.

The results of the confirmatory factor analysis in combination with the accepted levels of internal reliability offer the possibility to determine the respective scales, the calculation of the respective average values and their use in further analysis throughout the whole sample.

Most of the factors were positively correlated with each other (Table 10). The exception was the “Color” factor, which was unrelated to the “Protection and content recognition” and “Smart functionality” factors. Anyone who gave a high rating for one factor also gave a high rating to the other (especially among the factors INF, PHY2 and PHY3).

## 4. Conclusions

The original purpose of this paper was to demonstrate the multidisciplinary nature of food packaging. Researchers agree with older research results such as Coles, McDowell and Kirwan (2003) and Rundh [12], which support the need to consider many factors in the design of food product packaging. Factors such as promotion, safety, environmental impact and waste management of packaging material throughout the food life cycle, etc., are proposed.

The main research questions of this study are: RQ1: What do consumers consider the key elements of food packaging? RQ2: Can we group these elements into factors? To this end, primary data are collected through a questionnaire in order to identify the significance that consumers perceive against the various elements of the food product packaging. Researchers argue that by identifying the importance of key packaging elements, manufacturers in the food industry should include marketers, logistics experts, food scientists and environmental managers in the design and development of packaging in the food supply chain [36].

In particular, this paper proposed a tool that can detect a consumer’s attitude towards food packaging. The evaluation of the responses led to the identification of factors that predetermine a consumer’s attitude and the identification of the characteristics of the packaging that meet the consumer’s expectations. The results herein are directly applicable to business practices, especially in relation to the targeted promotion of a product on the market.

Exploratory factor analysis (EFA) was conducted on a random split-half sample of the data to examine the factor structure of these elements in the general population. Confirmatory factor analysis (CFA) was conducted in the holdout sample. The EFA of the packaging items resulted in a seven-factor solution: (1) informational content, (2) content protection and recognition, (3) smart functioning, (4) geometry, (5) environmental friendliness (6) endurance and (7) coloration. The CFA in the holdout sample supported this factor structure. The findings of the present study are informed by the consumer attitudes and predispositions towards packaging, thus having obvious managerial applications.

The research findings show that consumers recognize the important role that packaging plays in food safety and quality, and in relation to the information that it provides. It was proved that consumers understand and appreciate the multifunctional and multidisciplinary nature of packaging, considering that the informational, operational, physical and visual elements are of high importance. Specifically, research confirms that:(a)Health and nutrition are two interrelated concepts that receive constant attention from the consumers. This is aligned with recent studies for specific products such as of Hall et al. [37] that propose specific policies to restrict marketing and require health warnings on sugar-sweetened beverage packaging. Furthermore, Grummon, Taillie and Golden [38] argue that also sugar-sweetened beverage health warning policies could discourage sugar-sweetened beverage consumption. Many researchers identify the packaging information as a critical success factor for supporting health and nutrition. Küster-Boluda and Vila [39] confirm that the nutritional information and visual cues play a more relevant role than nutritional information response and informative cues.(b)The quality of a food product is inextricably linked to the quality of its packaging. This was also the result of a number of surveys. An indicative list of recent studies include Petrescu, Vermeir and Petrescu-Mag [40] who argue that the use frequency of food quality cues related to health is primarily influenced by the attention paid to food quality, and Crovato [41] who emphasize the usefulness of packaging information provision to increase the quality of the products.

It is also interesting to note that consumers have acquired a pro-environmental consciousness, which is in accordance with the study of Popovic, Bossink and van der Sijde [42], who examined the factors that influence the consumers’ decision to purchase food in environmentally friendly packaging. Pauer et al. [43], propose a methodological framework for environmental assessment of food packaging and Lehmann [44] proposes a platform for improving the understanding of what was really happening with, amongst other forms of waste, food at the consumer household level.

Limitations of the study include, first, the investigation of the significance of packaging elements without consideration of the various types/categories of food products. It is obvious that different perceptions reflect consumers with different needs and/or demographic characteristics. For example, when consumers look for packaged organic food products they select a product based on different criteria. Second, the study is focused on a part of the food supply chain. Opinions and appreciation of other key players of the food supply chain such as traders and intermediaries, wholesalers, retailers, and third-party logistics providers are of great importance.

Therefore, the findings from this research can be expanded by exploring the appreciation of the packaging elements from other roles/key players involved in the food supply chain such as executives who have different roles in the food industry (marketing, logistics and supply chain management, food technologist and managers that manage environmental initiatives and projects). The results of this research, combined with acceptable levels of internal reliability, offer the ability to define the corresponding scales, calculate the respective mean values and use this data in further statistical analyses of the entire sample.

## Figures and Tables

**Figure 1 foods-09-01097-f001:**
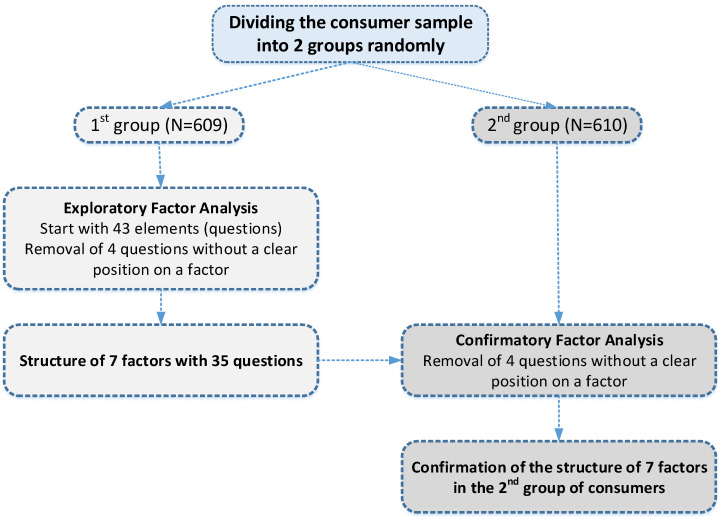
Research methodology steps.

**Figure 2 foods-09-01097-f002:**
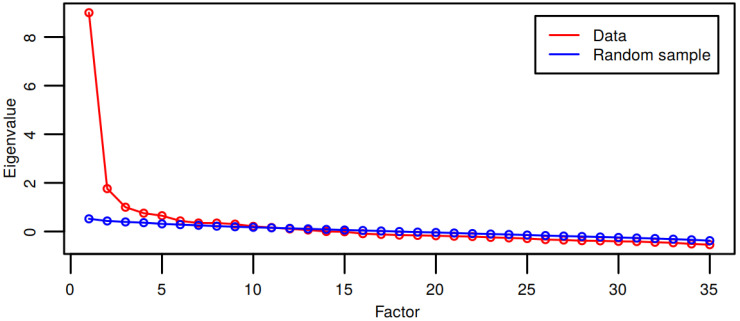
Comparison of a factor data structure with a random sample of the same size.

**Figure 3 foods-09-01097-f003:**
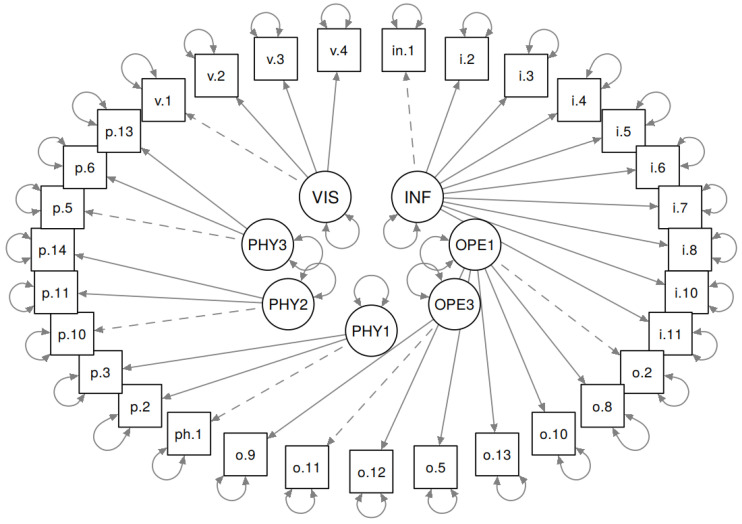
Latent factors and the elements they are reflected in (measurement model).

**Figure 4 foods-09-01097-f004:**
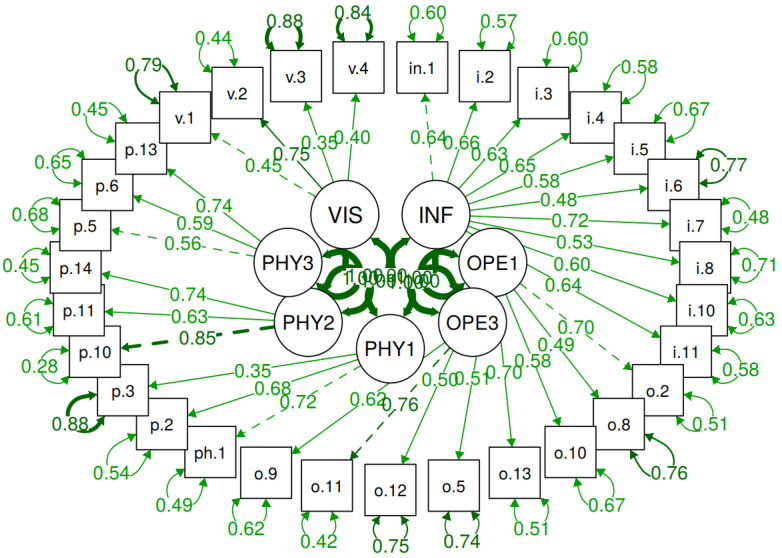
Standard model rates (consumers, *N* = 610).

**Table 1 foods-09-01097-t001:** Categorization of packaging elements for food products

Category	Packaging Elements
Informational elements	[Inf.q1] Provides nutrition information [Inf.q2] Reports production or reproduction techniques [Inf.q3] Includes quality standards marks [Inf.q4] Includes marks that show the compliance to environmental practices [Inf.q5] Includes data that support traceability [Inf.q6] Includes product identification coding schemes such as barcode, Quick Response (QR-code), etc. [Inf.q7] Includes marks for flammable / hazardous materials, storage conditions and brand elements [Inf.q8] Provides proposed ways of consumption [Inf.q9] Suggest recipes for this product [Inf.q10] Indicate country of origin and secondary materials [Inf.q11] Shows product temperature at any given time [Inf.q12] Designates protected origin name
Operational elements	[Oper.q1] Protects the product from theft [Oper.q2] Protects the product from moisture [Oper.q3] Can easily be mounted on the shelf [Oper.q4] Does not expose the product to light [Oper.q5] Allows visual contact with part of the product [Oper.q6] Can easily be transported and handled [Oper.q7] Allows packaging in larger packages/logistics units (carton, pallet, etc.) [Oper.q8] Is ready to cook [Oper.q9] Increases product life [Oper.q10] Has a smart label [Oper.q11] The shape of the package describes the product content [Oper.q12] Does not expose light to solar radiation
Physical elements	[Phys.q1] Has a marginally bigger size than the product size [Phys.q2] Has a marginally bigger volume than the product’s volume [Phys.q3] Follows common/typical shapes (e.g. square, rectangle, triangle, circle) [Phys.q4] Is made of durable materials [Phys.q5] Is waterproof [Phys.q6] Withstands mechanical stress [Phys.q7] Withstands corrosion and wear [Phys.q8] Is light/has low weight [Phys.q9] Is produced by material/components (e.g. is made of durable materials that add prestige to the product) [Phys.q10] Is produced by materials that are environmentally friendly [Phys.q11] Is produced by materials that can be reused and materials that allow for elongation [Phys.q12] Is cheap (low price of production or recycling) [Phys.q13] Does not allow odors to leak [Phys.q14] Is made of recycled materials
Visual elements	[Vis.q1] Has vivid / strong colors [Vis.q2] Has only one color (monochrome) [Vis.q3] Has only white color [Vis.q4] Has many blank parts (or has only white color as a background) [Vis.q5] Has a picture

**Table 2 foods-09-01097-t002:** Consumer’s demographics.

Demographics	Values	N (%)
Gender	Men	562 (46.1%)
	Women	657 (53.9%)
Place of residence	Large city	892 (73.2%)
	Small town	327 (26.8%)
Educational level	Tertiary	554, 45.5%)
	Postgraduate or doctoral degrees	275 (22.6%)
Age Category	16–25	245 (20.1%)
	26–35	295 (24.2%)
	36–45	344 (28.2%)
	46–55	231 (18.9%)
	>55	104 (8.5%)

**Table 3 foods-09-01097-t003:** Results of the exploratory structure analysis.

Exploratory Factor Analysis	Hierarchical Classification	MR1	MR6	MR3	MR5	MR9	MR2	MR7	MR4	MR8	MR10
phys.q11	PHYS2	0.7	0.08	0.01	0	−0.01	−0.01	−0.03	−0.09	0.05	−0.18
phys.q10	PHYS2	0.67	−0.01	0.07	0.01	0.09	0	0.12	0.08	−0.1	0.13
phys.q14	PHYS2	0.67	0.01	−0.02	0.03	0.01	−0.02	0	0.14	0.07	0.07
inf.q7	INF1	0.29	0.19	0.28	0	0.1	−0.06	0.05	0.18	−0.09	0.14
inf.q6	INF2	−0.07	0.64	0.02	−0.02	0.02	−0.03	0.02	0.17	0.01	−0.09
inf.q5	INF2	0.04	0.61	0.04	0.09	−0.01	0.08	0.11	−0.16	0.05	−0.06
inf.q2	INF1	0.09	0.48	−0.04	0.05	0.08	−0.1	−0.01	0.19	−0.05	0.21
inf.q4	INF1	0.3	0.45	−0.07	0.01	0.09	−0.02	0.08	−0.07	0.07	0.04
inf.q3	INF1	0.18	0.42	0.27	0	0.07	−0.03	−0.13	−0.09	0.01	0.19
inf.q1	INF1	0.15	0.26	0.05	0.03	0.11	−0.13	0.13	−0.16	0.12	0.24
phys.q7	PHYS4	−0.04	0.02	0.8	0.01	−0.06	0.03	0.13	0.02	0.02	−0.09
oper.q4	OPER1	0.08	−0.04	0.63	0.12	0.05	−0.01	−0.04	0	0.05	0.16
inf.q11	INF3	0.09	0.08	0.35	0.04	0.01	0.08	−0.05	0.03	0.27	0.25
inf.q8	INF3	0.04	0.08	0.34	0.11	0.2	−0.08	−0.1	0.06	0.19	−0.05
oper.q13	OPER1	−0.02	0.13	0.09	0.61	0.01	−0.02	−0.01	0.1	0.02	0.15
oper.q10	OPER2	0.08	0.03	0	0.5	−0.02	0.11	0.09	−0.01	0.15	−0.02
oper.q8	OPER3	−0.06	0.01	0.07	0.49	−0.03	0.06	0.03	0.18	0.07	−0.03
oper.q2	OPER1	0.09	0.05	0.25	0.49	0.1	−0.13	0.18	−0.08	−0.06	−0.03
phys.q8	PHYS1	0.07	−0.2	0.08	0.47	0.18	0.05	0.01	−0.03	0.14	−0.08
phys.q1	PHYS1	0.08	0.04	−0.01	−0.04	0.72	0	0.08	−0.08	0.13	0.02
phys.q2	PHYS1	−0.01	0.08	−0.09	0.37	0.56	0.06	−0.05	0.1	−0.07	−0.09
phys.q3	PHYS1	−0.02	−0.05	0.1	−0.1	0.44	0.27	0.06	0.2	0.05	0.1
vis.q4	VIS1	0.07	−0.01	0.06	0.05	−0.01	0.69	−0.01	−0.06	−0.07	0.07
vis.q2	VIS1	−0.14	0.03	−0.05	0.03	0.09	0.56	0.03	−0.03	0.07	−0.09
vis.q1	VIS2	0.09	−0.04	−0.04	−0.08	−0.04	0.35	−0.04	0.01	0.09	−0.29
vis.q3	VIS1	−0.07	0.04	0.05	−0.23	0.11	0.33	−0.01	0.23	0.02	0
phys.q5	PHYS3	−0.01	0.02	0.04	0	0.05	0	0.8	0.05	0.03	−0.04
phys.q13	PHYS3	0.15	−0.05	0.11	0.11	−0.11	−0.04	0.41	−0.08	0.09	0.34
inf.q10	INF3	0.11	0.19	0.01	0.08	−0.06	0.08	0.36	−0.06	0.04	0.26
oper.q11	OPER6	0.17	0.03	−0.05	0.17	−0.01	−0.03	0.04	0.5	0.17	0.11
phys.q6	PHYS4	0.08	0.11	0.19	0.07	0.04	−0.06	0.24	0.47	−0.07	−0.17
oper.q3	OPER6	0.12	−0.03	0.1	0.1	0.11	−0.1	0.14	0.45	0.1	0.01
oper.q9	OPER6	0.24	0.13	0.01	0.09	−0.18	0.1	0	0.34	0.27	−0.11
oper.q12	OPER4	−0.09	0	−0.01	0.06	0.08	0.05	0.12	0.1	0.56	0.1
oper.q5	OPER4	0.06	0.04	0.17	0.02	0.15	−0.1	0.04	−0.04	0.53	−0.12

**Table 4 foods-09-01097-t004:** Effectiveness and linear correlation of the proposed factors.

		MR1	MR6	MR3	MR5	MR9	MR2	MR7	MR4	MR8	MR10
Sums of Squared (SS) Loadings		2.55	2.17	2.26	2.2	1.58	1.3	1.61	1.43	1.37	0.92
Proportion Variance		0.07	0.06	0.06	0.06	0.05	0.04	0.05	0.04	0.04	0.03
Cumulative Variance		0.07	0.13	0.2	0.26	0.31	0.34	0.39	0.43	0.47	0.5
Proportion Explained		0.15	0.12	0.13	0.13	0.09	0.07	0.09	0.08	0.08	0.05
Cumulative Proportion		0.15	0.27	0.4	0.53	0.62	0.69	0.79	0.87	0.95	1
Factor Correlation	MR6	0.48									
	MR3	0.39	0.29								
	MR5	0.31	0.24	0.4							
	MR9	0.22	0.23	0.17	0.32						
	MR2	−0.15	−0.12	−0.06	−0.02	0.13					
	MR7	0.33	0.27	0.46	0.29	0.18	−0.06				
	MR4	0.21	0.17	0.17	0.25	0.16	0.03	0.21			
	MR8	0.28	0.18	0.31	0.3	0.26	0.11	0.27	0.18		
	MR10	0.3	0.16	0.21	0.1	0.03	−0.09	0.14	0	0.05	1

**Table 5 foods-09-01097-t005:** Correlation of factors proposed by the methods of hierarchical classification and the investigative structure analysis.

	MR1	MR2	MR3	MR4	MR5	MR6	MR7	MR8	MR9	MR10	Total
INF1	1					4					5
INF2						2					2
INF3			2				1				3
OPER1			1		2						3
OPER2					1						1
OPER3					1						1
OPER4								2			2
OPER6				3							3
PHYS1					1				3		4
PHYS2	3										3
PHYS3							2				2
PHYS4			1	1							2
VIS1		3									3
VIS2		1									1
Total	4	4	4	4	5	6	3	2	3	0	35

**Table 6 foods-09-01097-t006:** Final table with the questions/elements that did not participate in the grouping.

Question		Text
1	inf.q9	Suggests recipes for the product
2	oper.q1	Protects the product from theft
3	oper.q3	Can easily be mounted on the shelf
4	oper.q4	Does not expose the product to light
5	oper.q6	Can easily be transported and handled
6	oper.q7	Allows packaging of the product into larger units
7	phys.q4	Is made of durable materials
8	phys.q7	Withstands corrosion and wear
9	phys.q8	Is light/has a low weight
10	phys.q9	Is produced by materials that add prestige to the product
11	phys.q12	Is cheap
12	vis.q5	Has a picture

**Table 7 foods-09-01097-t007:** Structure of the questionnaire after the investigative structure analysis.

Items		Description of the Factor	Cronbach’s α
INF1	inf.q1, inf.q2, inf.q3, inf.q4	Quality characteristics of the product	0.768
INF2	inf.q5, inf.q6	Traceability	0.629
INF3	inf.q7, inf.q8	Various information	0.528
INF4	inf.q10, inf.q11	Country of origin	0.533
OPE1	oper.q2, oper.q8, oper.q10, oper.q13	Durability	0.743
OPE2	oper.q5, oper.q12	Recognition	0.591
OPE3	oper.q9, oper.q11	“Smart” functionality	0.622
PHY1	phys.q1, phys.q2, phys.q3	Geometric characteristics	0.689
PHY2	phys.q10, phys.q11, phys.q14	Environmentally friendly	0.772
PHY3	phys.q5, phys.q6, phys.q13	Durability	0.650
VIS	vis.q1, vis.q2, vis.q3, vis.q4	Coloration	0.566

**Table 8 foods-09-01097-t008:** Structure of the questionnaire after the consolidation of the factors with the non-accepted internal reliability.

Items	Text	Cronbach’s α
INF	Informational content	0.839
inf.q1	Provides nutritional information	
inf.q2	Provides information about the production techniques of the product	
inf.q3	Contains quality markings	
inf.q4	Includes markings indicating adherence to environmental practices	
inf.q5	Includes items that help make the product traceable	
inf.q6	Has traceability codes (barcode, Quick Response QR-code)	
inf.q7	Has markings to show if it is flammable or contains other dangerous materials	
inf.q8	Suggests methods of consumption of the product	
inf.q10	Indicates the country of origin of primary and auxiliary materials	
inf.q11	Indicates if a product is protected origin name	
OPE1	Protection and recognition of the product	0.757
oper.q2	Protects the product from humidity	
oper.q8	Is resistant to cooking/baking/roasting conditions etc.	
oper.q10	Increases the lifespan of the product	
oper.q13	Does not expose the product to solar radiation	
oper.q5	Allows visual contact with part of the product	
oper.q12	The shape of the packaging describes the contents of the product	
OPE2	“Smart” functionality	0.622
oper.q9	Shows the temperature of the product at any one time	
oper.q11	Is “smart”	
PHY1	Geometric characteristics	0.689
phys.q1	Has a size marginally larger than the size of the product	
phys.q2	Has a volume marginally larger than the volume of the product	
phys.q3	Is based on known geometric shapes	
PHY2	Environmentally friendly	0.772
phys.q10	Is made of materials that are environmentally friendly	
phys.q11	Is made of materials that can be reused	
phys.q14	Is made of recyclable materials	
PHY3	Durability	0.650
phys.q5	Is waterproof	
phys.q6	Tolerates mechanical vibrations	
phys.q13	Does not allow odors to leak out	
VIS	Coloration	0.566
vis.q1	Has bright colors	
vis.q2	Has only one color	
vis.q3	Has white color (background)	
vis.q4	Has blank spaces	

**Table 9 foods-09-01097-t009:** Results of the confirmatory structure analysis.

Adjustment Indicators	Methods	
	Multiple Linear Regression	Diagonally Weighted Least Squares (DWLS)
Repetitions	94	72
χ^2^	1940.0	1070.5
Df	413	413
χ^2^/df	4.7	2.6
NFI	0.737	0.944
CFI	0.779	0.965
GFI	0.826	0.970
AGFI	0.791	0.964
TLI	0.751	0.960
SRMR	0.066	0.063
RMSEA	0.078	0.051
95% Lower barrier	0.074	0.047
95% Upper barrier	0.081	0.055
*p* (RMSEA ≤ 0.05)	<0.001	0.306

**Table 10 foods-09-01097-t010:** Linear correlation between the factors (sample of consumers).

	INF	OPE1	OPE2	PHY1	PHY2	PHY3
OPE1	0.600 **					
OPE2	0.451 **	0.550 **				
PHY1	0.390 **	0.437 **	0.289 **			
PHY2	0.653 **	0.478 **	0.496 **	0.299 **		
PHY3	0.594 **	0.571 **	0.445 **	0.313 **	0.494 **	
VIS	−0.099 **	−0.001	0.000	0.211 **	−0.117 **	−0.083 **

** Statistically significant correlation at the 0.01 level.

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
