# Peer review of "Investigating Food Packaging Elements from a Consumer’s Perspective"

_foods, 2020, doi:10.3390/foods9081097_

Round 1

Reviewer 1 Report

This study identified and evaluated packaging elements in the food industry using a factor analysis. Though the survey and the statistical analysis were successfully conducted, I have some concerns primarily in regards to the scope and precondition of the study.

  • This study included the entire food packaging. However, the materials and formats of food packaging vary depending on types of food, biological/physical characteristics of food products, intended primary functions, particular regulations about the specific food, CPG type of food or bags for fresh produce, and etc. I assume that consumers' attitudes could be different depending on those conditions stated above. For example, the consumers' purchase behaviors on chocolate, fresh produce, chips, dairy food , and organic nutrition bars could be different. I am wondering if the authors considered those variables in this study, and how they dealt with them.
  • It will be great if authors more explicitly stated missing knowledge from the previous studies on this topic and the justification of the study in the introduction section. 

Author Response

Dear editor,
Below, you can find our revisions/responses regarding the meaningful and very useful comments and points of the reviewers.
Please, note that we have highlighted in yellow the revised parts.
Kindest regards,
Authors.

Reviewer 2 Report

Thank you for submitting your manuscript for the Special Issue “Food Packaging Strategies” in Foods. The topic analyzed in the article is very relevant and fits the special issue of the journal.

The sample size used in the study is especially noteworthy, which undoubtedly allows the results to be generalized to the Greek population.

However, I would like to point out certain aspects that have room for improvement:

1.- Added value of the paper: You should include wich is the added-value of your research in the Introduction Section. As the authors state in lines 98-10, “Overall, many researchers have tried to identify the key elements of the package (in general, and especially in the food industry) which have an impact in the four above-mentioned disciplines that are involved in the packaging of food products [24-28]”. In this sense, it is important that authors clarify the added value of their research.

2.- References. Food packaging is a topic that has been addressed by the most recent literature. In this sense, I invite the authors to make an update of the bibliographic sources used. In this way, the study will be better justified. For example:

- Ferreira, B. M. (2019). Packaging texture influences product taste and consumer satisfaction. Journal of Sensory Studies, 34(6), e12532.

- Crovato, S., Mascarello, G., Marcolin, S., Pinto, A., & Ravarotto, L. (2019). From purchase to consumption of bivalve molluscs: A qualitative study on consumers’ practices and risk perceptions. Food Control, 96, 410-420.

- Hall, M. G., Lazard, A. J., Grummon, A. H., Mendel, J. R., & Taillie, L. S. (2020). The impact of front-of-package claims, fruit images, and health warnings on consumers' perceptions of sugar-sweetened fruit drinks: Three randomized experiments. Preventive Medicine, 132, 105998.

- Küster-Boluda, I., & Vila, N. (2020). Can Health Perceptions, Credibility, and Physical Appearance of Low-Fat Foods Stimulate Buying Intentions?. Foods, 9(7), 866.

There are many studies published in the last two years that may be useful to authors.

3.- Objectives, research questions and/or hypotheses: Perhaps this is the aspect to which the authors should pay the greatest attention. This is a very high quality journal where only high quality academic papers are published. As the authors know, any scientific article should establish scientific objectives (research questions/hypotheses) based on the literature. In this regard, I encourage authors to do it based on the literature.

4.- Figures: please remember to use commas and dots and check your numbers.

I encourage the authors to review this article. My best wishes

Author Response

(The authors gave the same response as above.)

Reviewer 3 Report

  1. Author should include perspective towards the environmentally friendly packaging.
  2. In introduction, Author should consumer perceptions of packaging functions and material.
  3. Also, perspective towards packaging and environmental sustainability?
  4. Author must include more details about seven factor solution: (1) Informational content, (2) Content protection and recognition, (3) Smart functioning, (4) Geometry, (5) Environmentally friendly (6) Endurance and (7) Coloration

Author Response

(The authors gave the same response as above.)

Round 2

Reviewer 1 Report

Thanks for revising the introduction part adding the specific research questions and more reviews on the previous studies in regards to the topic.

However, I still have some concerns that the study generalized the results without a consideration of different types/categories of food packaging. Can the consumers purchasing behavior and perceptions on every food products always the same? (e.g., when consumers are looking for particular food products for their infants/children they might have different perceptions and select a product based on different criteria). I assume that the results could be influenced by those variables. Therefore, before the publication, I would like to recommend authors either to add a section to describe the limitations that could come from the variables or to justify the effect of those variables. 

Author Response

Thank you for the comments.
We have tried to incorporate them into the revised manuscript.

Reviewer 2 Report

Reference: foods-883220-review-v2

Thank you for submitting your manuscript for the Special Issue “Food Packaging Strategies” in Foods, and your improvements in the initial version.

Still I would like to remark two aspects:

1.- The authors state: “Furthermore, authors note that there is a lack of surveys about the main factors / components of packaging - which we call elements for marketing managers- from a holistic point of 66 view. That means that encompasses marketing, logistics, food technologists and corporate social responsibility (CSR) environmental issues. There are great research papers that emphasize the multidisciplinary nature of packaging and many researches regarding the importance of elements from (mainly) a marketing point of view and a small number of logistics/supply chain management point of view. Therefore, (we think) that there is a lack of strategies, approaches and tools for product designers, marketers and promoters, food technologists and process / service managers as well as engineers. This research effort aims to fill this gap or, at least, make the first important step. By recognizing the importance, marketing managers, logisticians, food technologists and corporate social responsibility (CSR) and environmental executives ascribe to it, and how they perceive it both in terms of packaging materials and retail packaging, producers can take into account those elements that are highly appreciated by all of the above components.”

But they do not support these arguments with literature. Please, provide cites. And please, use a more academic language, avoid using “authors note”, “we think”. Authors must make sentences based on literature, not in their supositions.

2.- Additionally, I still do not appreciate the added value of the paper. Authors point out two research questions: “The main research questions of this study are as follows: RQ1: What do consumers consider the key elements of food packaging? and

RQ2: Can we group these elements into factors?” but, in my opinion, previous literature has analysed and answered them. So, why this paper is relevant?

I encourage the authors to review this article. My best wishes

Author Response

(The authors gave the same response as above.)
